# A General Method for Amortizing Variational Filtering

**Joseph Marino, Milan Cvitkovic, Yisong Yue**
California Institute of Technology
{jmarino, mcvitkovic, yyue}@caltech.edu

## Abstract

We introduce the *variational filtering EM algorithm*, a simple, general-purpose method for performing variational inference in dynamical latent variable models using information from only past and present variables, i.e. filtering. The algorithm is derived from the variational objective in the filtering setting and consists of an optimization procedure at each time step. By performing each inference optimization procedure with an iterative amortized inference model, we obtain a computationally efficient implementation of the algorithm, which we call *amortized variational filtering*. We present experiments demonstrating that this general-purpose method improves performance across several deep dynamical latent variable models.

## 1 Introduction

Complex tasks with time-series data, like audio comprehension or robotic manipulation, must often be performed online, where the model can only consider past and present information. Models for such tasks, e.g. Hidden Markov Models, frequently operate by inferring the hidden state of the world at each time-step. This type of online inference procedure is known as *filtering*. Learning filtering models purely through supervised labels or rewards can be impractical, requiring massive collections of labeled data or significant efforts at reward shaping. In contrast, generative models can learn and infer hidden structure and states directly from data. Deep latent variable models [18, 27, 37], in particular, offer a promising direction; they infer latent representations using expressive deep networks, commonly using variational methods to perform inference [24]. Recent works have extended deep latent variable models to the time-series setting, e.g. [7, 12]. However, inference procedures for these dynamical models have been proposed on the basis of intuition rather than from a rigorous inference optimization perspective, potentially limiting performance.

We introduce *variational filtering EM*, an algorithm for performing filtering variational inference and learning that is rigorously derived from the variational objective. As detailed below, the variational objective in the filtering setting results in a sequence of inference optimization objectives, with one at each time-step. By initializing each of these inference optimization procedures from the corresponding prior distribution, a classic Bayesian prediction-update loop naturally emerges. This contrasts with existing filtering approaches for deep dynamical models, which use inference models that do not explicitly account for prior predictions during inference. However, using iterative inference models [32], which overcome this limitation, we develop a computationally efficient implementation of the variational filtering EM algorithm, which we refer to as *amortized variational filtering* (AVF).

The main contributions of this paper are the variational filtering EM algorithm and its amortized implementation, AVF. This general-purpose filtering algorithm is widely applicable to dynamical latent variable models, as we demonstrate in our experiments. Moreover, the variational filtering EM algorithm is derived from the filtering variational objective, providing a solid theoretical framework for filtering inference. By precisely specifying the inference optimization procedure, this method takes a simple form compared to previous hand–designed methods. Using several deep dynamical

latent variable models, we demonstrate that this filtering approach compares favorably against current methods across a variety of benchmark sequence data sets.

## 2 Background

Section 2.1 provides the general form of a dynamical latent variable model. Section 2.2 covers variational inference. Deep latent variable models are often trained efficiently by amortizing inference optimization (Section 2.3). Applying this technique to dynamical models is non-trivial, leading many prior works to use hand–designed amortized inference methods (Section 2.4).

### 2.1 Dynamical latent variable models

A sequence of $T$ observations, $\mathbf{x}_{\leq T}$, can be modeled using a dynamical latent variable model, $p_\theta(\mathbf{x}_{\leq T}, \mathbf{z}_{\leq T})$, which models the joint distribution between $\mathbf{x}_{\leq T}$ and a sequence of latent variables, $\mathbf{z}_{\leq T}$, with parameters $\theta$. It is typically assumed that $p_\theta(\mathbf{x}_{\leq T}, \mathbf{z}_{\leq T})$ can be factorized into conditional joint distributions at each step, $p_\theta(\mathbf{x}_t, \mathbf{z}_t | \mathbf{x}_{<t}, \mathbf{z}_{<t})$, which are conditioned on preceding variables. This results in the following auto-regressive formulation:

$$p_\theta(\mathbf{x}_{\leq T}, \mathbf{z}_{\leq T}) = \prod_{t=1}^{T} p_\theta(\mathbf{x}_t, \mathbf{z}_t | \mathbf{x}_{<t}, \mathbf{z}_{<t}) = \prod_{t=1}^{T} p_\theta(\mathbf{x}_t | \mathbf{x}_{<t}, \mathbf{z}_{\leq t}) p_\theta(\mathbf{z}_t | \mathbf{x}_{<t}, \mathbf{z}_{<t}). \tag{1}$$

$p_\theta(\mathbf{x}_t | \mathbf{x}_{<t}, \mathbf{z}_{\leq t})$ is the *observation* model, and $p_\theta(\mathbf{z}_t | \mathbf{x}_{<t}, \mathbf{z}_{<t})$ is the *dynamics* model, both of which can be arbitrary functions of their conditioning variables. However, while Eq. 1 provides the general form of a dynamical latent variable model, further assumptions about the dependency structure, e.g. Markov, or functional forms, e.g. linear, are often necessary for tractability.

### 2.2 Variational inference

Given a model and a set of observations, we typically want to *infer* the posterior for each sequence, $p(\mathbf{z}_{\leq T} | \mathbf{x}_{\leq T})$, and *learn* the model parameters, $\theta$. Inference can be performed online or offline through Bayesian filtering or smoothing respectively [38], and learning can be performed through maximum likelihood estimation. Unfortunately, inference and learning are intractable for all but the simplest model classes. For non-linear functions, which are present in *deep* latent variable models, we must resort to approximate inference. Variational inference [24] reformulates inference as optimization by introducing an approximate posterior, $q(\mathbf{z}_{\leq T} | \mathbf{x}_{\leq T})$, then minimizing the KL-divergence to the true posterior, $p(\mathbf{z}_{\leq T} | \mathbf{x}_{\leq T})$. To avoid evaluating $p(\mathbf{z}_{\leq T} | \mathbf{x}_{\leq T})$, one can express the KL-divergence as

$$D_{KL}(q(\mathbf{z}_{\leq T} | \mathbf{x}_{\leq T}) || p(\mathbf{z}_{\leq T} | \mathbf{x}_{\leq T})) = \log p_\theta(\mathbf{x}_{\leq T}) + \mathcal{F}, \tag{2}$$

where $\mathcal{F}$ is the *variational free energy*, also referred to as the (negative) *evidence lower bound* or ELBO, defined as

$$\mathcal{F} \equiv -\mathbb{E}_{q(\mathbf{z}_{\leq T} | \mathbf{x}_{\leq T})} \left[ \log \frac{p_\theta(\mathbf{x}_{\leq T}, \mathbf{z}_{\leq T})}{q(\mathbf{z}_{\leq T} | \mathbf{x}_{\leq T})} \right]. \tag{3}$$

In Eq. 2, $\log p_\theta(\mathbf{x}_{\leq T})$ is independent of $q(\mathbf{z}_{\leq T} | \mathbf{x}_{\leq T})$, so one can minimize the KL-divergence to the true posterior, thereby performing approximate inference, by minimizing $\mathcal{F}$ w.r.t. $q(\mathbf{z}_{\leq T} | \mathbf{x}_{\leq T})$. Further, as KL-divergence is non-negative, Eq. 2 implies that free energy upper bounds the negative log likelihood. Therefore, upon minimizing $\mathcal{F}$ w.r.t. $q(\mathbf{z}_{\leq T} | \mathbf{x}_{\leq T})$, one can use the gradient $\nabla_\theta \mathcal{F}$ to learn the model parameters. These two optimization procedures are respectively the expectation and maximization steps of the variational EM algorithm [34], which alternate until convergence. To scale this algorithm, stochastic gradients can be used for both inference [35] and learning [21].

### 2.3 Amortized variational inference

Performing inference optimization using conventional stochastic gradient descent techniques can be computationally demanding, potentially requiring many inference iterations. To increase efficiency, a separate inference model can learn to map data examples to approximate posterior estimates [8, 18, 27, 37], thereby amortizing inference across examples [16]. Denoting the distribution parameters of $q$ as $\boldsymbol{\lambda}^q$ (e.g. Gaussian mean and variance), standard inference models take the form

$$\boldsymbol{\lambda}^q \leftarrow f_\phi(\mathbf{x}), \tag{4}$$

where the inference model is denoted as $f$ with parameters $\phi$. These models, though efficient, have limitations. Notably, because these models only receive the data as input, they are unable to account for empirical priors, which occur from one latent variable to another. Such priors arise in the dynamics of dynamical models, forming priors across time steps, as well as in hierarchical models, forming priors across levels. Previous works have neglected to include empirical priors during inference, attempting to overcome this limitation through heuristics, like "top-down" inference in hierarchical models [40] and recurrent inference models in dynamical models, e.g. [7].

Iterative inference models [32] directly account for these priors, instead performing inference optimization by iteratively encoding approximate posterior estimates and gradients:

$$\boldsymbol{\lambda}^q \leftarrow f_\phi(\boldsymbol{\lambda}^q, \nabla_{\boldsymbol{\lambda}^q} \mathcal{F}). \tag{5}$$

The gradients, $\nabla_{\boldsymbol{\lambda}^q} \mathcal{F}$, can be estimated through black box methods [35] or the reparameterization trick [27, 37] when applicable. Analogously to learning to learn [1], iterative inference models learn to perform inference optimization, thereby *learning to infer*. Eq. 5 provides a viable encoding form for an iterative inference model, but other forms, such as additionally encoding the data, $\mathbf{x}$, can potentially lead to faster inference convergence. Empirically, iterative inference models have also been shown to yield improved modeling performance over comparable standard models [32].

## 2.4    Related work

Many deterministic deep dynamical latent variable models have been proposed for sequential data [6, 41, 30, 10]. While these models often capture many aspects of the data, they cannot account for the uncertainty inherent in many domains, typically arising from partial observability of the environment. By averaging over multi-modal distributions, these models often produce samples in regions of low probability, e.g. blurry video frames. This inadequacy necessitates moving to probabilistic models, which can explicitly model uncertainty to accurately capture the distribution of possible sequences.

Amortized variational inference [27, 37] has enabled many recently proposed probabilistic deep dynamical latent variable models, with applications to video [42, 26, 43, 23, 15, 11, 3, 9, 29, 19], speech [7, 12, 17, 22, 29], handwriting [7], music [12], etc. While these models differ in their functional mappings, most fall within the general form of Eq. 1. Crucially, simply encoding the observation at each step is insufficient to accurately perform approximate inference, as the prior can vary across steps. Thus, with each model, a hand-crafted amortized inference procedure has been proposed. For instance, many filtering inference methods re-use various components of the generative model [7, 12, 15, 9], while some methods introduce separate recurrent neural networks into the filtering procedure [4, 9] or encode the previous latent sample [26]. Specifying a filtering method has been an engineering effort, as we have lacked a theoretical framework.

The variational filtering EM algorithm precisely specifies the inference optimization procedure implied by the filtering variational objective. The main insight from this analysis is that, having drawn approximate posterior samples at previous steps, inference becomes a local optimization, depending only on the current prior and observation. This suggests one unified approach that explicitly performs inference optimization at each step, replacing the current collection of custom filtering methods. When the approximate posterior at each step is initialized at the corresponding prior, this approach entails a Bayesian prediction-update loop, with the update composed of a gradient (error) signal.

Perhaps the closest technique in the probabilistic modeling literature is the "residual" inference method from Fraccaro *et al.* [12], which updates the approximate posterior mean from the prior. Similar ideas have been proposed on an empirical basis for deterministic models [30, 20]. PredNet [30] is a deterministic model that encodes prediction errors to perform inference. This approach is inspired by predictive coding [36, 13], a theory from neuroscience that postulates that feedforward pathways in sensory processing areas of the brain use prediction errors to update state estimates from prior predictions. In turn, this theory is motivated by classical Bayesian filtering [38], which updates the posterior from the prior using the likelihood of the prediction. For linear Gaussian models, this manifests as the Kalman filter [25], which uses prediction errors to perform exact inference.

Finally, several recent works have used particle filtering in conjunction with amortized inference to provide a tighter lower bound on the log likelihood for sequential data [31, 33, 28]. The techniques developed here can also be applied to this tighter bound.

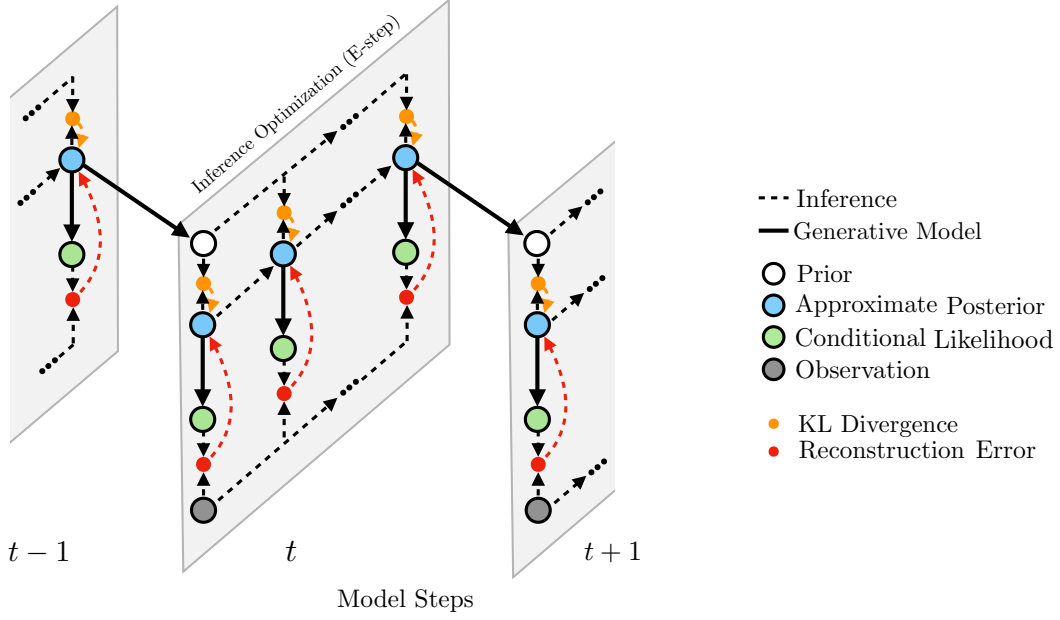

Figure 1: **Variational filtering EM.** The diagram shows filtering inference within a dynamical latent variable model, as outlined in Algorithm 1. The central gray region depicts inference optimization of the approximate posterior, $q(\mathbf{z}_t|\mathbf{x}_{\leq t}, \mathbf{z}_{<t})$, at step $t$, which can be initialized at or near the corresponding prior, $p_\theta(\mathbf{z}_t|\mathbf{x}_{<t}, \mathbf{z}_{<t})$. Sampling from the approximate posterior generates the conditional likelihood, $p_\theta(\mathbf{x}_t|\mathbf{x}_{<t}, \mathbf{z}_{\leq t})$, which is evaluated at the observation, $\mathbf{x}_t$, to calculate the reconstruction error. This term is combined with the KL divergence between the approximate posterior and prior, yielding the step free energy, $\mathcal{F}_t$ (Eq. 9). Inference optimization (E-step) involves finding the approximate posterior that minimizes the step free energy terms. Learning (M-step), which is not shown, corresponds to updating the model parameters, $\theta$, to minimize the total free energy, $\mathcal{F}$.

## 3 Variational filtering

Section 3.1 describes variational filtering EM (Algorithm 1), a general algorithm for performing filtering variational inference in dynamical latent variable models. In Section 3.2, we introduce a method for amortizing inference optimization using iterative inference models.

### 3.1 Variational filtering expectation maximization (EM)

In the filtering setting, the approximate posterior at each step is conditioned only on information from past and present variables, enabling *online* approximate inference. This implies a structured approximate posterior, in which $q(\mathbf{z}_{\leq T}|\mathbf{x}_{\leq T})$ factorizes across steps as

$$q(\mathbf{z}_{\leq T}|\mathbf{x}_{\leq T}) = \prod_{t=1}^{T} q(\mathbf{z}_t|\mathbf{x}_{\leq t}, \mathbf{z}_{<t}). \tag{6}$$

Note that the conditioning variables in each term of $q$ denote an *indirect* dependence that arises through free energy minimization and does not necessarily constitute a direct functional mapping. Under a filtering approximate posterior, the free energy (Eq. 3) can be expressed as

$$\mathcal{F} = \sum_{t=1}^{T} \mathbb{E}_{\prod_{\tau=1}^{t-1} q(\mathbf{z}_\tau|\mathbf{x}_{\leq \tau}, \mathbf{z}_{<\tau})} \left[ \mathcal{F}_t \right] = \sum_{t=1}^{T} \tilde{\mathcal{F}}_t, \tag{7}$$

(see Appendix A for the derivation) where $\mathcal{F}_t$ is the *step free energy*, defined as

$$\mathcal{F}_t \equiv -\mathbb{E}_{q(\mathbf{z}_t|\mathbf{x}_{\leq t}, \mathbf{z}_{<t})} \left[ \log \frac{p_\theta(\mathbf{x}_t, \mathbf{z}_t|\mathbf{x}_{<t}, \mathbf{z}_{<t})}{q(\mathbf{z}_t|\mathbf{x}_{\leq t}, \mathbf{z}_{<t})} \right], \tag{8}$$

**Algorithm 1** Variational Filtering Expectation Maximization

---
1: **Input:** observation sequence $\mathbf{x}_{1:T}$, model $p_\theta(\mathbf{x}_{1:T}, \mathbf{z}_{1:T})$
2: $\nabla_\theta \mathcal{F} = 0$        ▷ parameter gradient
3: **for** $t = 1$ **to** $T$ **do**
4:      initialize $q(\mathbf{z}_t | \mathbf{x}_{\leq t}, \mathbf{z}_{<t})$        ▷ at/near $p_\theta(\mathbf{z}_t | \mathbf{x}_{<t}, \mathbf{z}_{<t})$
5:      $\tilde{\mathcal{F}}_t := \mathbb{E}_{q(\mathbf{z}_{<t} | \mathbf{x}_{<t}, \mathbf{z}_{<t-1})} [\mathcal{F}_t]$
6:      $q(\mathbf{z}_t | \mathbf{x}_{\leq t}, \mathbf{z}_{<t}) = \arg\min_q \tilde{\mathcal{F}}_t$        ▷ inference (E-Step)
7:      $\nabla_\theta \mathcal{F} = \nabla_\theta \mathcal{F} + \nabla_\theta \tilde{\mathcal{F}}_t$
8: **end for**
9: $\theta = \theta - \alpha \nabla_\theta \mathcal{F}$        ▷ learning (M-Step)

---

and we have also defined $\tilde{\mathcal{F}}_t$ as the $t^{\text{th}}$ term in the summation. Note that with a single step, the filtering free energy reduces to the first step free energy, thereby recovering the static case. As in this setting, the step free energy can be re-expressed as a reconstruction term and a KL-divergence term:

$$\mathcal{F}_t = -\mathbb{E}_{q(\mathbf{z}_t | \mathbf{x}_{\leq t}, \mathbf{z}_{<t})} [\log p_\theta(\mathbf{x}_t | \mathbf{x}_{<t}, \mathbf{z}_{\leq t})] + D_{KL}(q(\mathbf{z}_t | \mathbf{x}_{\leq t}, \mathbf{z}_{<t}) || p_\theta(\mathbf{z}_t | \mathbf{x}_{<t}, \mathbf{z}_{<t})). \quad (9)$$

The filtering free energy in Eq. 7 is the sum of these step free energy terms, each of which is evaluated according to expectations over past latent sequences. To perform filtering variational inference, we must find the set of $T$ terms in $q(\mathbf{z}_{\leq T} | \mathbf{x}_{\leq T})$ that minimize the filtering free energy summation.

We now describe the variational filtering EM algorithm, given in Algorithm 1 and depicted in Figure 1, which optimizes Eq. 7. This algorithm sequentially optimizes each of the approximate posterior terms to perform filtering inference. Consider the approximate posterior at step $t$, $q(\mathbf{z}_t | \mathbf{x}_{\leq t}, \mathbf{z}_{<t})$. This term appears in $\mathcal{F}$, either directly or in expectations, in terms $t$ through $T$ of the summation:

$$\mathcal{F} = \underbrace{\tilde{\mathcal{F}}_1 + \tilde{\mathcal{F}}_2 + \cdots + \tilde{\mathcal{F}}_{t-1} + \overbrace{\tilde{\mathcal{F}}_t}^{\text{terms in which } q(\mathbf{z}_t | \mathbf{x}_{\leq t}, \mathbf{z}_{<t}) \text{ appears}}}_{\text{steps on which } q(\mathbf{z}_t | \mathbf{x}_{\leq t}, \mathbf{z}_{<t}) \text{ depends}} + \tilde{\mathcal{F}}_{t+1} + \cdots + \tilde{\mathcal{F}}_{T-1} + \tilde{\mathcal{F}}_T. \quad (10)$$

However, the filtering setting dictates that the optimization of the approximate posterior at each step can only condition on past and present variables, i.e. steps 1 through $t$. Therefore, of the $T$ terms in $\mathcal{F}$, the *only* term through which we can optimize $q(\mathbf{z}_t | \mathbf{x}_{\leq t}, \mathbf{z}_{<t})$ is the $t^{\text{th}}$ term:

$$q^*(\mathbf{z}_t | \mathbf{x}_{\leq t}, \mathbf{z}_{<t}) = \underset{q(\mathbf{z}_t | \mathbf{x}_{\leq t}, \mathbf{z}_{<t})}{\arg\min} \tilde{\mathcal{F}}_t. \quad (11)$$

Optimizing $\tilde{\mathcal{F}}_t$ requires evaluating expectations over previous approximate posteriors. Again, because approximate posterior estimates cannot be influenced by future variables, these past expectations remain *fixed* through the future. Thus, variational filtering (the variational E-step) can be performed by sequentially minimizing each $\mathcal{F}_t$ w.r.t. $q(\mathbf{z}_t | \mathbf{x}_{\leq t}, \mathbf{z}_{<t})$, holding the expectations over past variables fixed. Conveniently, once the past expectations have been evaluated, inference optimization is entirely defined by the free energy at that step.

For simple models, such as linear Gaussian models, these expectations may be computed exactly. However, in general, the expectations must be estimated through Monte Carlo samples from $q$, with inference optimization carried out using stochastic gradients [35]. As in the static setting, we can initialize $q(\mathbf{z}_t | \mathbf{x}_{\leq t}, \mathbf{z}_{<t})$ at (or near) the prior, $p_\theta(\mathbf{z}_t | \mathbf{x}_{<t}, \mathbf{z}_{<t})$. This yields a simple interpretation: starting with $q$ at the prior, we generate a *prediction* of the data through the likelihood, $p_\theta(\mathbf{x}_t | \mathbf{x}_{<t}, \mathbf{z}_{\leq t})$, to evaluate the current step free energy. Using the approximate posterior gradient, we then perform an inference *update* to the estimate of $q$. This resembles classical Bayesian filtering, where the posterior is updated from the prior prediction according to the likelihood of observations. Unlike the classical setting, reconstruction and update steps are repeated until inference convergence.

After inferring an optimal approximate posterior, learning (the variational M-step) can be performed by minimizing the total filtering free energy w.r.t. the model parameters, $\theta$. As Eq. 7 is a summation and differentiation is a linear operation, $\nabla_\theta \mathcal{F}$ is the sum of contributions from each of these terms:

$$\nabla_\theta \mathcal{F} = \sum_{t=1}^{T} \nabla_\theta \left[ \mathbb{E}_{\prod_{\tau=1}^{t-1} q(\mathbf{z}_\tau | \mathbf{x}_{\leq \tau}, \mathbf{z}_{<\tau})} [\mathcal{F}_t] \right]. \quad (12)$$

Parameter gradients can be estimated online by accumulating the result from each term in the filtering free energy. The parameters are then updated at the end of the sequence. For large data sets, stochastic estimates of parameter gradients can be obtained from a mini-batch of data examples [21].

## 3.2  Amortized variational filtering

Performing approximate inference optimization (Algorithm 1, Line 6) with traditional techniques can be computationally costly, requiring many iterations of gradient updates and hand-tuning of optimizer hyper-parameters. In online settings, with large models and data sets, this may be impractical. An alternative approach is to employ an amortized inference model, which can learn to minimize $\mathcal{F}_t$ w.r.t. $q(\mathbf{z}_t|\mathbf{x}_{\leq t}, \mathbf{z}_{<t})$ more efficiently at each step. Note that $\mathcal{F}_t$ (Eq. 8) contains $p_\theta(\mathbf{x}_t, \mathbf{z}_t|\mathbf{x}_{<t}, \mathbf{z}_{<t}) = p_\theta(\mathbf{x}_t|\mathbf{x}_{<t}, \mathbf{z}_{\leq t})p_\theta(\mathbf{z}_t|\mathbf{x}_{<t}, \mathbf{z}_{<t})$. The prior, $p_\theta(\mathbf{z}_t|\mathbf{x}_{<t}, \mathbf{z}_{<t})$, varies across steps, constituting the latent dynamics. Standard inference models, which only encode $\mathbf{x}_t$, do not have access to the prior and therefore cannot properly optimize $q(\mathbf{z}_t|\mathbf{x}_{\leq t}, \mathbf{z}_{<t})$. Many inference models in the sequential setting attempt to account for this information by including hidden states, e.g. [7, 12, 9]. However, given the complexities of many generative models, it can be difficult to determine how to properly route the necessary prior information into the inference model. As a result, each dynamical latent latent variable model has been proposed with an accompanying custom inference model set-up.

We propose a simple and general alternative method for amortizing filtering inference that is agnostic to the particular form of the generative model. Iterative inference models [32] naturally account for the changing prior through the approximate posterior gradients. These models are thus a natural candidate for performing inference at each step. Similar to Eq. 5, when $q(\mathbf{z}_t|\mathbf{x}_{\leq t}, \mathbf{z}_{<t})$ is a parametric distribution with parameters $\boldsymbol{\lambda}_t^q$, the inference update takes the form:

$$\boldsymbol{\lambda}_t^q \leftarrow f_\phi(\boldsymbol{\lambda}_t^q, \nabla_{\boldsymbol{\lambda}_t^q}\tilde{\mathcal{F}}_t). \tag{13}$$

We refer to this set-up as *amortized variational filtering* (AVF). As in Eq. 5, we note that Eq. 13 offers just one particular encoding form for an iterative inference model. For instance, $\mathbf{x}_t$ could be additionally encoded at each step. Marino *et al.* also note that in latent Gaussian models, precision-weighted errors provide an alternative inference optimization signal [32]. There are two main benefits to using iterative inference models in the filtering setting:

- The approximate posterior is updated from the prior, so model capacity is utilized for inference *corrections* rather than re-estimating the approximate posterior at each step.

- These inference models contain all of the terms necessary to perform inference optimization, providing a simple model form that does not require any additional hidden states or inputs.

In practice, these advantages permit the use of relatively simple iterative inference models that can perform filtering inference efficiently and accurately. We demonstrate this in the following section.

## 4  Experiments

We empirically evaluate amortized variational filtering using multiple deep dynamical latent Gaussian model architectures on a variety of sequence data sets. Specifically, we use AVF to train VRNN [7], SRNN [12], and SVG [9] on speech [14], music [5], and video [39] data. In each setting, we compare AVF against the originally proposed filtering method for the model. Diagrams of the filtering methods are shown in Figure 2. Implementations of the models are based on code provided by the respective authors of VRNN[1], SRNN[2], and SVG[3]. Accompanying code can be found online at `github.com/joelouismarino/amortized-variational-filtering`.

### 4.1  Experiment set-up

Iterative inference models are implemented as specified in Eq. 13, encoding the approximate posterior parameters and their gradients at each inference iteration at each step. Following [32], we normalize the inputs to the inference model using layer normalization [2]. The generative models that we

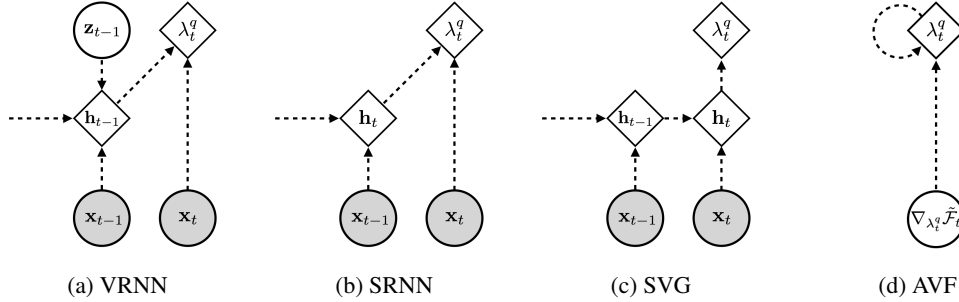

(a) VRNN     (b) SRNN     (c) SVG     (d) AVF

Figure 2: Filtering inference models for VRNN, SRNN, SVG, and AVF. Each diagram shows the computational graph for inferring the approximate posterior parameters, $\boldsymbol{\lambda}^q$, at step $t$. Previously proposed methods rely on hand-crafted architectures of observations, hidden states, and latent variables. AVF is a simple, general filtering procedure that only requires the local inference gradient.

evaluate contain non-spatial latent variables, thus, we use fully-connected layers to parameterize the inference models. Importantly, minimal effort went into engineering the inference model architectures: across all models and data sets, we utilize the *same* inference model architecture for AVF. Further details are found in Appendix B.

### 4.1.1 Speech modeling

**Models** For speech modeling, we use VRNN and SRNN, attempting to keep the model architectures consistent with the original implementations. The most notable difference in our implementation occurs in SRNN, where we use an LSTM rather than a GRU as the recurrent module. As in [12], we anneal the KL divergence initially during training. In both models, we use a Gaussian output density. Unlike [7, 12, 17], which evaluate log *densities*, we evaluate and report log *probabilities* by integrating the output density over the data discretization window, as in modeling image pixels. This permits comparison across different output distributions.

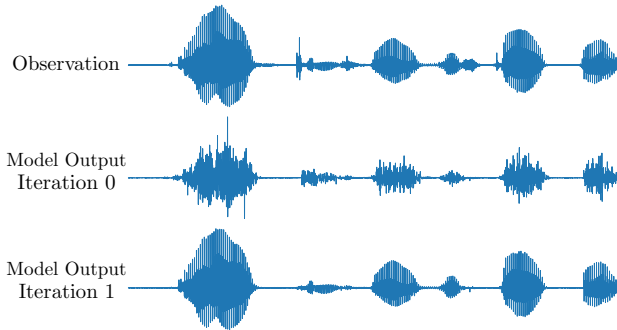

Figure 3: Test data (top), output predictions (middle), and reconstructions (bottom) for TIMIT using SRNN with AVF. Sequences run from left to right. The predictions made by the model already contain the general structure of the data. AVF explicitly updates the approximate posterior from the prior prediction, focusing on inference *corrections* rather than re-estimation.

**Data** We train and evaluate on TIMIT [14], which consists of audio recordings of 6,300 sentences spoken by 630 individuals. As performed in [7], we sample the audio waveforms at 16 kHz, split the training and validation sets into half second clips, and group each sequence into bins of 200 consecutive samples. Thus, each training and validation sequence consists of 40 model steps. Evaluation is performed on the full duration of each test sequence, averaging roughly 3 seconds.

### 4.1.2 Music modeling

**Model** We model polyphonic music using SRNN. The generative model architecture is the same as in the speech modeling experiments, with changes in the number of layers and units to match [12]. To model the binary music notes, we use a Bernoulli output distribution. Again, we anneal the KL divergence initially during training.

**Data** We use four data sets of polyphonic (MIDI) music [5]: Piano-midi.de, MuseData, JSB Chorales, and Nottingham. Each data set contains between 100 and 1,000 songs, with each song

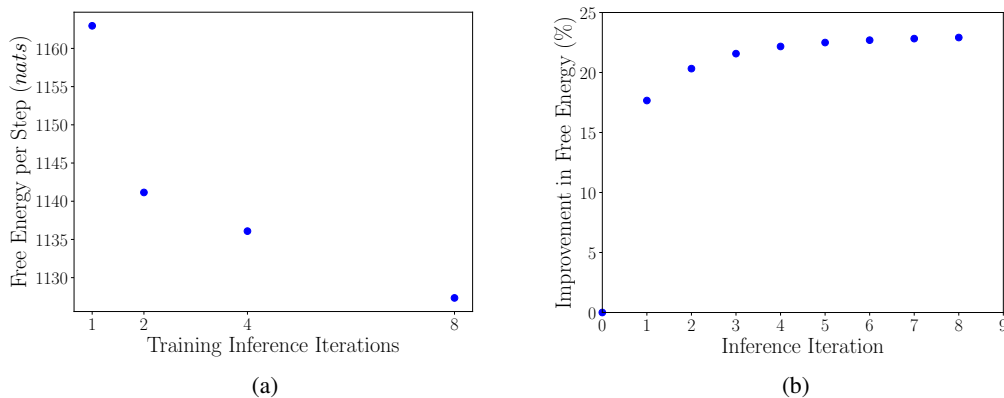

<div align="center">(a)                  (b)</div>

Figure 4: **Improvement with inference iterations.** Results are shown on the TIMIT validation set using VRNN with AVF. (a) Average free energy per step with varying numbers of inference iterations during training. Additional iterations tend to result in improved performance. (b) Average relative improvement in free energy from the initial (prior) estimate at each inference iteration for a single model. Empirically, each successive iteration provides further, smaller improvements.

between 100 to 4,000 steps. For training and validation, we break the sequences into clips of length 25, and we test on the entire test sequences.

### 4.1.3 Video modeling

**Model** Our implementation of SVG differs from the original model in that we evaluate conditional log-likelihood under a Gaussian output density rather than mean squared output error. All other architecture details are identical to the original model. However, [9] down-weight the KL-divergence by a factor of 1e-6 at all steps. We instead remove this factor to use the free energy during training and evaluation. As to be expected, this results in the model using the latent variables to a lesser extent. We train and evaluate SVG using filtering inference at all steps, rather than predicting multiple steps into the future, as in [9].

**Data** We train and evaluate SVG on KTH Actions [39], which contains 760 train / 768 val / 863 test videos of people performing various actions, each of which is between roughly 50 - 150 frames. Frames are re-sized to $64 \times 64$. For training and validation, we split the data into clips of 20 frames.

## 4.2 Results

### 4.2.1 Additional Inference Iterations

The variational filtering EM algorithm involves inference optimization at each step (Algorithm 1, Line 6). AVF optimizes each approximate posterior through a model that learns to perform iterative updates (Eq. 13). Additional inference iterations may lead to further improvement in performance [32]. We explore this aspect on TIMIT using VRNN. In Figure 4a, we plot the average free energy per step on validation sequences for models trained with varying numbers of inference iterations. Figure 4b shows average relative improvement over the prior estimate for a single model trained with 8 inference iterations. We observe that training with additional inference iterations empirically leads to improved performance (Figure 4a), with each iteration providing diminishing improvement during inference (Figure 4b). This aspect is distinct from many baseline filtering methods, which directly output the approximate posterior at each step.

We can also directly visualize inference improvement through the model output. Figure 3 illustrates example reconstructions over inference iterations, using SRNN on TIMIT. At the initial inference iteration, the approximate posterior is initialized from the prior, resulting in an output prediction. The iterative inference model then uses the approximate posterior gradients to update the estimate, improving the output reconstruction.

Table 1: Average free energy per step (in *nats*) on the TIMIT speech data set for SRNN and VRNN with the respective originally proposed filtering procedures (baselines) and with AVF.

|         |    | TIMIT |
|---------|----|-------|
| VRNN    |    |       |
| baseline |   | **1,082** |
| AVF     |    | 1,105 |
| SRNN    |    |       |
| baseline |   | 1,026 |
| AVF     |    | **1,024** |

Table 2: Average free energy per step (in *nats*) on the KTH Actions video data set for SVG with the originally proposed filtering procedure (baseline) and with AVF.

|          |    | KTH Actions |
|----------|----|-------------|
| SVG      |    |             |
| baseline |   | 15,097      |
| AVF      |    | **11, 714** |

Table 3: Average free energy per step (in *nats*) on polyphonic music data sets for SRNN with and without AVF. Results from Fraccaro *et al.* [12] are provided for comparison, however, our model implementation differs in several aspects (see Appendix B).

|               |    | Piano-midi.de | MuseData | JSB Chorales | Nottingham |
|---------------|----|---------------|----------|--------------|------------|
| SRNN          |    |               |          |              |            |
| baseline [12] |   | 8.20          | 6.28     | 4.74         | 2.94       |
| baseline      |   | 8.19          | 6.27     | **6.92**     | 3.19       |
| AVF           |   | **8.12**      | **5.99** | 6.97         | **3.13**   |

### 4.2.2 Quantitative Comparison

Tables 1, 2, and 3 present quantitative comparisons of average filtering free energy per step between AVF (with 1 inference iteration per step) and baseline filtering methods for TIMIT, KTH Actions, and the polyphonic music data sets respectively. On TIMIT, training with AVF performs comparably to the baseline methods for both VRNN and SRNN. We note that VRNN with AVF using 2 inference iterations resulted in a final test performance of 1,071 *nats* per step, outperforming the baseline method. Similar results are also observed on each of the polyphonic music data sets. Again, increasing the number of inference iterations to 5 for AVF on JSB Chorales resulted in a final test performance of 6.77 *nats* per step. AVF significantly improves the performance of SVG on KTH Actions. We attribute this, likely, to the absence of the KL down-weighting factor in our training objective as compared with [9]. The baseline filtering procedure seems to struggle to a greater degree than AVF. From comparing the results above, we see that AVF is a general filtering procedure that performs well across multiple models and data sets, despite using a relatively simple inference model structure.

## 5   Conclusion

We introduced the variational filtering EM algorithm for filtering in dynamical latent variable models. Variational filtering inference can be expressed as a sequence of optimization objectives, linked across steps through previous latent samples. Using iterative inference models to perform inference optimization, we arrived at an efficient implementation of the algorithm: amortized variational filtering. This general filtering algorithm scales to large models and data sets. Numerous methods have been proposed for filtering in deep dynamical latent variable models, with each method hand–designed for each model. The variational filtering EM algorithm provides a single framework for analyzing and constructing these methods. Amortized variational filtering is a simple, theoretically-motivated, and general filtering method that we have shown performs on-par with or better than multiple existing state-of-the-art methods.

#### Acknowledgments

We would like to thank Matteo Ruggero Ronchi for helpful discussions. This work was supported by the following grants: JPL PDF 1584398, NSF 1564330, and NSF 1637598.

## Footnotes

[1] `https://github.com/jych/nips2015_vrnn`

[2] `https://github.com/marcofraccaro/srnn`

[3] `https://github.com/edenton/svg`

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
