[Supplementary Material · amortized_variational_filtering_supplementary-2.pdf]

## A Filtering variational free energy

This derivation largely follows that of [6] and is valid for factorized filtering approximate posteriors. From Eq. 3, we have the definition of variational free-energy:

$$\mathcal{F} \equiv -\mathbb{E}_{q(\mathbf{z}_{\leq T}|\mathbf{x}_{\leq T})}\left[\log \frac{p_\theta(\mathbf{x}_{\leq T}, \mathbf{z}_{\leq T})}{q(\mathbf{z}_{\leq T}|\mathbf{x}_{\leq T})}\right]. \tag{1}$$

Plugging in the forms of the joint distribution (Eq. 1) and approximate posterior (Eq. 6), we can write the term within the expectation as a sum:

$$\mathcal{F} = -\mathbb{E}_{q(\mathbf{z}_{\leq T}|\mathbf{x}_{\leq T})}\left[\log\left(\prod_{t=1}^{T} \frac{p(\mathbf{x}_t, \mathbf{z}_t|\mathbf{x}_{<t}, \mathbf{z}_{<t})}{q(\mathbf{z}_t|\mathbf{x}_{\leq t}, \mathbf{z}_{<t})}\right)\right] \tag{2}$$

$$\mathcal{F} = -\mathbb{E}_{q(\mathbf{z}_{\leq T}|\mathbf{x}_{\leq T})}\left[\sum_{t=1}^{T} \log \frac{p(\mathbf{x}_t, \mathbf{z}_t|\mathbf{x}_{<t}, \mathbf{z}_{<t})}{q(\mathbf{z}_t|\mathbf{x}_{\leq t}, \mathbf{z}_{<t})}\right] \tag{3}$$

$$\mathcal{F} = -\mathbb{E}_{q(\mathbf{z}_{\leq T}|\mathbf{x}_{\leq T})}\left[\sum_{t=1}^{T} C_t\right] \tag{4}$$

where the term $C_t$ is defined to simplify notation. We then expand the expectation:

$$\mathcal{F} = -\mathbb{E}_{q(\mathbf{z}_1|\mathbf{x}_1)}\cdots\mathbb{E}_{q(\mathbf{z}_T|\mathbf{x}_{\leq T}, \mathbf{z}_{<T})}\left[\sum_{t=1}^{T} C_t\right] \tag{5}$$

There are $T$ terms within the sum, but each $C_t$ only depends on the expectations up to time $t$ because we only condition on past and present variables. This allows us to write:

$$\begin{aligned}\mathcal{F} = &-\mathbb{E}_{q(\mathbf{z}_1|\mathbf{x}_1)}\left[C_1\right]\\ &-\mathbb{E}_{q(\mathbf{z}_1|\mathbf{x}_1)}\mathbb{E}_{q(\mathbf{z}_2|\mathbf{x}_{\leq 2}, \mathbf{z}_1)}\left[C_2\right]\\ &-\cdots\\ &-\mathbb{E}_{q(\mathbf{z}_1|\mathbf{x}_1)}\mathbb{E}_{q(\mathbf{z}_2|\mathbf{x}_{\leq 2}, \mathbf{z}_1)}\cdots\mathbb{E}_{q(\mathbf{z}_T|, \mathbf{x}_{\leq T}, \mathbf{z}_{<T})}\left[C_T\right]\end{aligned} \tag{6}$$

$$\mathcal{F} = -\sum_{t=1}^{T}\mathbb{E}_{q(\mathbf{z}_{\leq t}|\mathbf{x}_{\leq t})}\left[C_t\right] \tag{7}$$

$$\mathcal{F} = -\sum_{t=1}^{T}\mathbb{E}_{\prod_{\tau=1}^{t} q(\mathbf{z}_\tau|\mathbf{x}_{\leq\tau}, \mathbf{z}_{<\tau})}\left[C_t\right] \tag{8}$$

$$\mathcal{F} = -\sum_{t=1}^{T}\mathbb{E}_{\prod_{\tau=1}^{t-1} q(\mathbf{z}_\tau|\mathbf{x}_{\leq\tau}, \mathbf{z}_{<\tau})}\left[\mathbb{E}_{q(\mathbf{z}_t|\mathbf{x}_{\leq t}, \mathbf{z}_{<t})}\left[C_t\right]\right] \tag{9}$$

As in Section 3, we define $\mathcal{F}_t$ as

$$\mathcal{F}_t \equiv -\mathbb{E}_{q(\mathbf{z}_t|\mathbf{x}_{\leq t}, \mathbf{z}_{<t})}\left[C_t\right] \tag{10}$$

$$\mathcal{F}_t = -\mathbb{E}_{q(\mathbf{z}_t|\mathbf{x}_{\leq t}, \mathbf{z}_{<t})}\left[\log \frac{p_\theta(\mathbf{x}_t, \mathbf{z}_t|\mathbf{x}_{<t}, \mathbf{z}_{<t})}{q(\mathbf{z}_t|\mathbf{x}_{\leq t}, \mathbf{z}_{<t})}\right]. \tag{11}$$

This allows us to write Eq. 9 as

$$\mathcal{F} = \sum_{t=1}^{T}\mathbb{E}_{\prod_{\tau=1}^{t-1} q(\mathbf{z}_\tau|\mathbf{x}_{\leq\tau}, \mathbf{z}_{<\tau})}\left[\mathcal{F}_t\right], \tag{12}$$

which agrees with Eq. 7.

# B   Implementation details

For all iterative inference models, we followed [7], using two layer fully-connected networks with 1,024 units per layer, "highway" gating connections [8], and ELU non-linearities [3]. Unless otherwise noted, these models received a concatenation of the current estimate of the approximate posterior distribution parameters and the corresponding gradients (4 terms in total), normalizing each term separately using layer normalization [1]. We used the output gating update employed in [7]. We also found that applying layer normalization to the approximate posterior mean estimates resulted in improved training stability. AVF is comparable in runtime to the baseline filtering methods. For example, with our implementation of SRNN on TIMIT, AVF requires 13.1 ms per time step, whereas the baseline method requires 15.6 ms per time step. AVF requires an additional decoding step, but inference is local to the current step, meaning that backpropagation is more efficient. Accompanying code can be found online at `github.com/joelouismarino/amortized-variational-filtering`.

## B.1   Speech modeling

The VRNN architecture is implemented as in [2], matching the number of layers and units in each component of the model, as well as the non-linearities. We trained on TIMIT with sequences of length 40, using a batch size of 64. For the baseline method, we used a learning rate of 0.001, as specified in [2]. For AVF, we used a learning rate of 0.0001. We annealed the learning rates by a factor of 0.999 after each epoch. Both models were trained for 700 epochs.

We implement SRNN following [5], with the exception of an LSTM in place of the GRU. All other architecture details, are kept consistent, including the use of clipped ($\pm 3$) leaky ReLU non-linearities. The sequence length and batch size are the same as above. We use a learning rate of 0.001 for the baseline method, following [5]. We use a learning rate of 0.0001. We use the same learning rate annealing strategy as above. Following [5], we anneal the KL-divergence of the baseline linearly over the first 20 epochs. We increase this duration to 50 epochs for AVF to account for the lower learning rate. The iterative inference model additionally encodes the data observation at each step, which we found necessary to overcome the local minima from the KL-divergence. Models were trained for 1,000 epochs.

## B.2   Music modeling

Our SRNN implementation is the same as in the speech modeling setting, with the appropriate changes in the number of units and layers to match [5]. We used a sequence length of 25 and a batch size of 16. All models were trained with a learning rate of 0.0001, with a decay factor of 0.999 per epoch. We annealed the KL-divergence linearly over the first 50 epochs. Models were trained for 800 epochs.

## B.3   Video modeling

The SVG model architecture is implemented identically to [4], with an additional log-variance term in the observation model's output to provide a Gaussian output density. Unlike [4], we do not down-weight the KL-divergence term in the free energy in order to achieve a proper bound. We note that this is likely the reason that our SVG baseline performs poorly, as the original model was not intended to be trained with the true variational bound objective. Indeed, we observed that the model struggles to make use of the latent variables, even when annealing the KL-divergence weight in the objective. We trained on sequences of length 20 using a batch size of 20. For both methods, we used a learning rate of 0.0001, with decay of 0.999 after each epoch. Models were trained for 100 epochs.