[Reviews · NeurIPS 2018]

Reviewer 1



Thank you for your thorough response. I appreciate the additional analysis and I think it will definitely make for a stronger paper. I also appreciate the run time discussion --- I was confused about the computational costs of the amortization, and the author response cleared that up for me. I am still a bit confused about why AVF would be less susceptible to local minima (and more likely to use the latent codes) than the baseline comparisons, and it would be great to include some discussion about this improvement. I have increased my score. ------------------ The authors present a method for fitting dynamic latent variable models with a variational inference objective. They use a variational family factorized sequentially to approximate the posterior over the latent variables. They describe the inference procedure using this posterior sequence approximation, which involves solving T optimization problems at each step of the sequence, approximating filtering. Learning the global parameters requires performing this as an inner loop. To reduce computation, they use iterative inference models as an amortized inference scheme. They present experimental comparisons to baseline methods in a variety of domains involving sequence modeling. *Clarity*: The presentation is relatively clear --- the introduction clearly states the problem to be solved and approach they take to solve it. Some aspects were confusing, particularly equations 5 and 13 (and further details of iterative inference mode). *Quality*: The paper appears to be technically correct, and the proposed method is tested in a variety of domains. The idea of using a sequentially structured inference network is interesting. I think the results section would be more compelling if alternate inference network schemes were compared (or if a baseline contains an amortization strategy, how does the structure compare?). *Originality+Impact*: The approach showed big empirical gains in some real data examples. I think the potential impact of this method is contingent on some practical aspects, such as run time. *Questions and Comments*: - I am a little confused about the term empirical priors, as used in line 82. Inference models of the form in Eq (4) have difficulty with which particular distribution? Is the idea that dynamical priors (or even hierarchical priors) induce some structure that is ignored by a generic inference network? Is the type of structure that the amortized filtering inference approach exploits? - Iterative inference models are a big part of the proposed method, however they are not well described in this manuscript. Could some part of section 2.3 explain Eq (5) in greater detail. - I am a bit confused about the units used in the experiment log-likelihood reporting. Do the experiments report the probability of test sequences (having marginalized out the latent variables)? - How does optimization differ between the using the existing filtering methods vs AVF? Is it a matter of optimization (e.g. would the existing baselines given more time or tuning rise to the performance of AVF? Is AVF uniformly better? - For the video modeling --- what is driving this big improvement? Is the posterior approximation more accurate? In general, does the use of the AVF framework enable more expressive posterior approximations than the baselines? Or does it afford more efficient inference? Or both? - How much does the amortization afford you? If the same variational approximation were used within a stochastic variational inference framework (say for a small dataset), we would expect it to outperform an amortized framework in terms of ELBO objective value. How does including the amortization structure affect the final model fit? - For learning global parameters, would a filter + smoothing approach be sensible? Can this approach be used to update global parameters using information from the entire sequence to do learning?

Reviewer 2



Summary This paper revisits variational filtering by incorporating iterative inference models, i.e. amortization models of the form \lambda_{t+1} = f(\lambda_{t}, \nabla F) where \lambda_{t+1} are the new variational parameters, \lambda_{t} are the old ones, and \nabla F is the gradient of the objective. As the filtering setting simplifies the multi-step objective into a one-step variational free energy (Equation 11), the paper proposes updating the generative parameters by (1) running one or more steps of an iterative inference model to minimize the local variational posterior at each time step, (2) compute the gradient of the one-step free energy, and (3) aggregate these gradients across time steps. Experiments are reported comparing the proposed variational filtering approach to three previously proposed inference methods each having multi-step dependencies. Speech, video, and music datasets are used for the experiments. Pros I found this paper interesting, informative, novel, and useful. Using iterative inference models to simplify the classific filtering problem is a useful methodological contribution that is shown to result in experimental improvements on three very different datasets. I also found the paper to be clear in its exposition; I especially like the diagrams in Figures 1 and 2. Lastly, the paper’s references are extensive and clearly explains the works context (Section 2.4). Cons One important question the paper does not address to my satisfaction is: why do we expect this local optimization procedure to perform better than inference approaches that account for multiple steps (such as the others presented in Figure 2)? And if in theory we don’t expect it to, why does it out-perform the other methods in practice? Is it that other approaches ‘waste parameters’ in that they must learn each local posterior from scratch (i.e. similar to the top-down inference approaches)? Or is it that the proposed approach’s initialization from the prior simplifies optimization? What happens if the iterative inference is given a ‘dumb’ initialization? Evaluation This paper presents a simple yet effective inference strategy that is validated over several data sets. Therefore I recommend acceptance. Although, I think the paper could be improved by analysis / simulation studies of the inference mechanics and not just the model’s final predictive performance.

Reviewer 3



This work introduces a new method for inference in deep dynamical latent Gaussian models. When working with the filtering assumption during variational learning, the current time-step's [t] posterior estimates are independent of the entire past and depend only on the variational parameters found in the previous time-step [t-1]. To calculate the approximate posterior distribution for the current time step [q_t], this work holds fixed q_{t-1} and performs local updates (by maximizing the ELBO) to find the best q_t possible for the current time step. This is used for inference and parameter estimation. The idea is straightforward and follows upon related recent work in the static (non-temporal) setting. e.g. http://www.people.fas.harvard.edu/~yoonkim/data/sa-vae.pdf http://proceedings.mlr.press/v80/marino18a.html https://arxiv.org/abs/1801.03558 https://arxiv.org/abs/1710.06085 The paper is generally clearly written and Figure 1 was useful to understanding the high level idea. I like the idea, its simple and does appear to work well in some situations. That said, I think the paper could be improved with some additional detail: * How is the argmin in Equation (11) solved? In the experimental section, I felt the paper jumped straight to the benchmarks without pause to explore any of the algorithmic contributions in detail: * What happens as the number of inference steps is increased? Did only a single step of inference yield the best results? * It would be useful to provide some intuition on where this approach helps -- one way to achieve this might to categorize which time steps see the most improvement in likelihood from improving the fidelity of the filtering based variational parameters.